# PCNA Unloading Is Crucial for the Bypass of DNA Lesions Using Homologous Recombination

**DOI:** 10.3390/ijms25063359

**Published:** 2024-03-15

**Authors:** Matan Arbel-Groissman, Batia Liefshitz, Nir Katz, Maxim Kuryachiy, Martin Kupiec

**Affiliations:** The Shmunis School of Biomedicine and Cancer Research, The George S. Wise Faculty of Life Sciences, Tel Aviv University, Tel Aviv 69978, Israel; matanarble@gmail.com (M.A.-G.);

**Keywords:** *Saccharomyces cerevisiae*, DNA repair, DNA damage bypass, post-replicational repair, PCNA, Elg1, Srs2, DNA replication, ubiquitin, SUMO

## Abstract

DNA Damage Tolerance (DDT) mechanisms allow cells to bypass lesions in the DNA during replication. This allows the cells to progress normally through the cell cycle in the face of abnormalities in their DNA. PCNA, a homotrimeric sliding clamp complex, plays a central role in the coordination of various processes during DNA replication, including the choice of mechanism used during DNA damage bypass. Mono-or poly-ubiquitination of PCNA facilitates an error-prone or an error-free bypass mechanism, respectively. In contrast, SUMOylation recruits the Srs2 helicase, which prevents local homologous recombination. The Elg1 RFC-like complex plays an important role in unloading PCNA from the chromatin. We analyze the interaction of mutations that destabilize PCNA with mutations in the Elg1 clamp unloader and the Srs2 helicase. Our results suggest that, in addition to its role as a coordinator of bypass mechanisms, the very presence of PCNA on the chromatin prevents homologous recombination, even in the absence of the Srs2 helicase. Thus, PCNA unloading seems to be a pre-requisite for recombinational repair.

## 1. Introduction

During the cell cycle, the DNA faces numerous attacks from exogenous or endogenous sources. Damaged DNA, if left untreated, can cause stalling or even collapse of the replication fork. A variety of DNA repair mechanisms exist in order to repair the damaged DNA and additional mechanisms are able to bypass lesions during DNA replication, allowing their repair in a post-replicative manner [1]. Many of the DNA repair and damage bypass mechanisms, which can collectively be called DNA Damage Tolerance (DDT) pathways, are orchestrated by post-translational modifications of the PCNA clamp [2,3]. PCNA is a homotrimeric complex, composed in the yeast of three identical Pol30 proteins, which serves as a sliding clamp and processivity factor for the different DNA polymerases [4,5]. PCNA is modified by a variety of post-translational modifications (PTMs), including ubiquitination, SUMOylation, and phosphorylation. These PTMs regulate the activity of PCNA and its interaction with many proteins involved in DNA replication and DDT; most PTMs are tightly evolutionarily conserved [6].

In response to DNA damage, PCNA is modified by ubiquitination at lysine 164 [7,8]. This ubiquitination, carried out by the Rad6–Rad18 complex, signals for the recruitment of DNA damage tolerance (DDT) proteins to the damaged site. Whereas mono-ubiquitination activates an error-prone DNA damage bypass orchestrated in the yeast by Pol-Zeta [9,10], further poly-ubiquitination of PCNA by Rad5-Mms2-Ubc13 facilitates a template switch bypass of the lesion in an error-free manner [11,12,13].

PCNA is loaded on the chromatin by the replication factor C complex (RFC) composed of the Rfc1-5 proteins, and is unloaded by a complex which shares the four small subunits Rfc2-5 with RFC, but contains Elg1 instead of Rfc1 (the Elg1 RFC-like complex or RLC) [14,15]. The timely unloading of PCNA by the Elg1 RLC plays a major role in replication and DNA damage bypass and repair [16]. Defects in Elg1 (ATAD5 in mammals) function lead to increased chromosome instability, elevated recombination and mutation rates, and additional genomic instability phenotypes [17]. In the absence of Elg1, PCNA, and in particular SUMOylated PCNA, accumulate on the chromatin [18], suggesting that SUMOylation of PCNA may be a signal for its recruitment [19]. PCNA undergoes SUMOylation at lysines 164 and 127 [3].

Elg1 has a complicated genetic interaction with Srs2, another protein that is recruited to PCNA following its SUMOylation [20]. Srs2 is a DNA helicase that plays a critical role in the regulation of DNA damage repair and bypass pathways in budding yeast [21,22,23]. Following DNA damage, ssDNA may be created, which is bound by RPA and later replaced by Rad51, forming a nucleoprotein filament [24]. This filament can interfere with DNA replication and transcription [25]. The Srs2 helicase unwinds the Rad51 filament and removes it from the DNA. This activity is necessary during the repair of double-stranded breaks [23,26], but also prevents the formation of unwanted recombination intermediates and allows the cell to repair the DNA damage by other mechanisms [27,28]. Srs2 is recruited to PCNA following its SUMOylation at lysines 164 or 127 [22]. By evicting Rad51 from the DNA, it down-regulates the alternative, homologous recombination-based “salvage pathway” [29].

In this paper, we show that PCNA plays a role, not only in recruiting and coordinating DNA damage repair proteins, but also that its presence on DNA, even in the absence of Srs2, can serve to prevent recombination in certain situations. By enclosing DNA, PCNA may sometimes serve as a physical constraint against recombinational repair.

## 2. Results

### 2.1. Spontaneous Unloading of PCNA Suppresses the Sensitivity Conferred by an Inactive DDT Mechanism

Because Pol30 is an essential protein, it is not possible to study its function by deleting the gene that encodes it. To overcome this limitation, we made use of Pol30 mutants that form a PCNA complex that is prone to spontaneously disassemble and fall from the chromatin [30,31]. This allows us to measure how the cells react to low levels of PCNA on the chromatin. Numerous DPP (Disassembly Prone PCNA) mutants were characterized. We focused on two mutations, E143K and V180D, which were shown to lower the affinity of the Pol30 subunits to each other. Without showing differences in the level of expression of the *POL30* gene, *pol30-E143K* mutants were shown in the past to exhibit a mild—and in the case of *pol30-V180D,* a strong—reduction in the level of PCNA on the chromatin [30]. We confirmed these results in our genetic background (Figure 1).

Lysine 164 of Pol30 is the main residue of PCNA that undergoes ubiquitination and SUMOylation [9]. The ubiquitination of K164 is a prerequisite for the activation of most of the DDT pathways [3], and as such, mutations that replace this lysine greatly sensitize the cells to DNA damaging agents as both the error prone and error free repair pathways are inactive [9]. It was previously noted that, perhaps counter-intuitively, an additional mutation in residue K127 suppresses the sensitive K164R phenotype [9]. This is due to the fact that SUMOylation at lysine 127 can still recruit Srs2, even in a strain mutated for lysine 164. By evicting Rad51 from the DNA, Srs2 down-regulates the salvage pathway (which is functional in the single *pol30-K164R* mutant). In a double mutant (*pol30-K127R*, *K164R*) Srs2 is not recruited, and the salvage pathway remains open, providing repair to the damaged DNA and reducing the sensitivity [29].

Surprisingly, the DPP mutations are able to suppress the DNA damage sensitivity of *pol30-K164R* strains (Figure 2). Adding the *pol30-V180D* mutation restores the *pol30-K164R* mutant to the same level of sensitivity conferred by the *pol30-K127R* mutation, whereas the suppression using *pol30-E143K* is slightly weaker, but clear (Figure 2). Given the stronger effect of *pol30-V180D*, we continued to characterize this mutation in particular.

### 2.2. Unstable PCNA Results in Low Amount of Srs2 on the Chromatin

The most logical explanation of the fact that DPP mutations suppress the sensitivity of *pol30-K164R* is that the reduced amount of PCNA on the chromatin de-facto lowered the amount of Srs2 on the chromatin, thus activating the salvage pathway. The hypothesis arising from this model is that the suppression will be similar in level to that provided by *pol30-K127R* and the deletion of *SRS2* will be epistatic to both. Figure 3 shows that this is indeed the case. The *pol30-K164R*, *K127R* and *pol30-K164R*, *V180D*, and even the triple *pol30-K164R*, *K127R*, *V180D* mutant, show the same sensitivity to MMS, whether the Srs2 helicase is active or completely absent (Figure 3).

To further validate this claim, we checked the level of Srs2 protein on the chromatin in the different mutants (Figure 4). As expected, mutating lysine 164 of PCNA decreases the amount of Srs2 on the chromatin, consistent with the role of lysine 164 as the main SUMOylation site of the Pol30 protein. The DPP mutation *pol30-V180D* on its own reduces Srs2 levels on the chromatin even more than *pol30-K164R*. Importantly, when the two mutations are combined, the levels of Srs2 on the chromatin are almost undetectable, similarly to what is observed in a *pol30-K164R*, *K127R* double mutant that lacks PCNA SUMOylation altogether (Figure 4).

### 2.3. DPP Suppression of pol30-K164R Is Dependent on Homologous Recombination (HR)

If, as hinted by the results from the previous section, the DPP mutation indeed activates the salvage pathway by lowering the amount of Srs2 on the chromatin, this suppression should be dependent on HR proteins [32]. Most HR reactions in the cell depend on the activity of Rad52 [33,34]. As *rad52*∆ cells are extremely sensitive to MMS, we turned to UV, a DNA damaging agent to which they are relatively resilient, allowing us to check the relationship between the gene deletions. In Figure 5, we show that whereas neither *RAD52* deletion nor any of the PCNA mutations render the strain sensitive to the inflicted UV levels, combining *rad52*∆ with either *pol30-K164R*, *K127R,* or with *pol30-K164R*, *V180D* greatly sensitizes the cells. We believe that this is because both double mutants are unable to recruit Srs2 to the fork. In the absence of ubiquitination at lysine 164, which prevents the opening of the DDT, cells use the salvage pathway, which is now open due to the lack of Srs2 [29]. Without Rad52, however, this mechanism fails, resulting in greatly increased sensitivity to UV. These results demonstrate that, similarly to the suppression of *pol30-K164R* by *pol30-K127R* [22], the suppression by *pol30-V180D* is due to the activity in the salvage pathway. Interestingly, deletion of *RAD52* causes a mild increase in the sensitivity of *pol30-V180D*; this is possibly because the spontaneous unloading of PCNA in this mutant precludes the use of DDT pathways and thus, results in an increased dependency on the salvage pathway. At the same time, the difference in sensitivity of the double, when compared to the single *rad52*∆ mutant, implies that PCNA plays an important role in the repair of UV damage when HR is not an option.

### 2.4. Suppression of the Synthetic Sickness of elg1∆srs2∆ Double Mutants by Spontaneous Unloading of PCNA

Whereas singly deleting either the *ELG1* or the *SRS2* genes does not result in a great increase in DNA damage sensitivity, the *elg1*∆ *srs2*∆ double mutant is extremely sick and exhibits great sensitivity to DNA damage [35]. This can be due to the fact that PCNA may act as a constraint, preventing HR. In the absence of Srs2, the salvage pathway may be activated, but the increased levels of PCNA on chromatin (due to lack of clamp unloading in the absence of Elg1) may curtail its performance. Figure 6A shows that the extreme DNA damage sensitivity of *elg1*∆ *srs2*∆ double mutants can be completely rescued by the *pol30-V180D* mutation, implying that spontaneous unloading of PCNA suppresses the DNA damage sensitivity of the strain, thus strengthening our claim. Moreover, the *pol30-V180D* mutation is also able to suppress the synthetic sickness of the double mutant strain (Figure 6B), again supporting the claim that PCNA serves as a physical constraint against homologous recombination.

## 3. Discussion

Despite the important role of the damage bypass mechanisms in cell survival and in the prevention of cancer, as well as their role in evolution, their regulation remains poorly understood. One of the more interesting, but puzzling, points about the DDT pathways is the role of Srs2 and its activity in the salvage pathway. The presence of Srs2 at the fork, recruited by the SUMOylation of PCNA, actively discourages recombination by preventing the formation of Rad51 filaments on DNA [27], a prerequisite for most homologous recombination (HR) events [33]. This, in turn, allows for the activation of the DDT pathways, either mutagenic or error free. When Srs2 is absent from the fork, the DNA lesion can be bypassed using HR, circumventing the need for any of the other DDT pathways or proteins [36]. PCNA ubiquitination at lysine 164 acts as a master activation switch for the DDT pathways. Therefore, mutations that prevent that ubiquitination confer extreme sensitivity to DNA damage [22]. Preventing the recruitment of Srs2 by deleting the gene, or mutating the second lysine that can undergo SUMOylation (K127) partially suppresses the sensitivity. Here, we have shown that similar results can be obtained by introducing mutations that cause spontaneous disassembly of PCNA, such as *pol30-V180D* or *pol30-K143D*, which reduce the amount of PCNA on chromatin (Figure 2 and Figure 3).

Our results support a model in which the very presence of PCNA on the chromatin constrains the ability of the cells to bypass lesions using the recombinational salvage pathway. The fact that it is enough to delete *SRS2*, bypassing the sensitivity of any other mutation in the DDT pathways, implies that when Srs2 is not there, all repair or bypass is canalized towards HR. For example, deletion of any of the genes in the error-free repair pathway group usually results in a greatly elevated mutation rate, as now all repair is being sent to the error-prone repair pathways. However, when combined with Srs2 deletion, the elevated mutation rate is suppressed [36,37]. This demonstrates again that the lack of Srs2 from the fork results in the uncontrolled activation of the salvage recombination and under these circumstances all the other DDT pathways remain unused. So, why is the deletion of *ELG1* so harmful to strains lacking Srs2? The most plausible explanation, as previously stated, is that once the process of HR has started, the inability of cells lacking Elg1 to unload PCNA results in recombination intermediates that are left unresolved, leading to cell toxicity and death. Figure 7 shows a schematic model of the bypass of a DNA lesion in a Srs2-deficient strain in a post-replicative manner. PCNA timely unloading and reloading is important for allowing the strand exchange, as PCNA encloses a DNA duplex in which one of the strands is damaged, but repair requires an exchange. PCNA should be unloaded and further reloaded since the bypass requires copying the sister DNA duplex and thus, the enclosing of the newly synthesized strand and the complementary one from the sister chromatid.

The importance of PCNA unloading in salvage recombination pathway explains why a Pol30 mutant that spontaneously disassembles from the chromatin can suppress the synthetic sickness of *elg1*∆ *srs2*∆ strains. When the salvage pathway is continually activated (as is the case in *srs2*∆ strains), PCNA has to be timely unloaded or it will lead to fork arrest. An alternative explanation, although less probable in our opinion, is that there is a third partner in this dance: a mysterious protein that binds to PCNA and, when mounted on the chromatin, causes havoc and damage to the cell. This harmful protein would normally compete with Srs2 to bind with PCNA and could also be usually regulated by the unloading of PCNA by Elg1. When both Elg1 and Srs2 are deleted, the lack of competition for binding to PCNA and the elevated levels of chromatin bound PCNA results in a high amount of the toxic protein on the chromatin, leading to the synthetic sickness and extreme sensitivity. We believe that such a protein would have already caught the attention of the scientific community, although this remains a formal possibility. Future genetic screens may confirm or negate this possibility.

Srs2 and Elg1 have complicated roles to perform throughout the cell cycle. Whereas Srs2 moves along with the fork during the S phase [20] preventing local recombination events, Elg1 is also continuously needed at the fork to cycle PCNA, at each Okazaki fragment during the replication of the lagging strand [38]. They both have SUMO-interacting motifs, and exhibit a preference for SUMOylated PCNA. Several important questions remain: How is Srs2 unloaded? Does the Elg1 RLC normally unload it together with the SUMOylated PCNA? Do Srs2 and Elg1 compete for the binding to SUMOylated PCNA? A future avenue to pursue is understanding the temporal and spatial regulation of Srs2 presence and activity. As it currently stands, almost all our knowledge of this pathway is derived from experiments that include an artificial removal of Srs2 from the fork. When, and how, is Srs2 removed from the fork under normal circumstances? For what purposes? This, we believe is the next major hurdle in the challenging research involving this protein.

## 4. Materials and Methods

Unless differently stated, all strains are derivatives of E134, and share the following genotype: ade5-1,lys2::InsEa14,trp1-289,his7-2,leu2-3,112,ura3-52 [39]. Table 1 shows all strains used in this study.

Standard yeast media and methods were used. Standard yeast molecular biology techniques were used to create the mutant collection.

### 4.1. Chromatin Fractionation Assay

Cells from 50 mL cultures (OD600 < 1.0) were collected by centrifugation, successively washed with ddH2O, PSB (20 mM Tris–Cl pH 7.4, 2 mM EDTA, 100 mM NaCl, 10 mMb-ME), and SB (1 M Sorbitol, 20 mM Tris–Cl pH 7.4), and transferred to a 2 mL Eppendorf tube. Cells were suspended in 1 mL SB, 30 ul Zymolase 20T (20 mg/mL in SB) was added, and samples were incubated at 30 °C with rotation until >85% spheroplasts were observed (60–90 min). Spheroplasts were collected by centrifugation (2 K, 5 min, 4 °C), washed twice with SB, and suspended in 500 mL EBX (20 mM Tris–Cl pH 7.4, 100 mM NaCl, 0.25% Triton X-100,15 mM-ME + protease/phosphatase inhibitors). TritonX-100 was added to a final concentration of 0.5% to lyse the outer cell membrane, and the samples were kept on ice for 10 min with gentle mixing. The lysate was layered over 1 mL NIB (20 mM Tris–Cl pH 7.4, 100 mM NaCl, 1.2 M sucrose, 15 mM-ME + protease/phosphatase inhibitors) and centrifuged at 12 K RPM for 15 min at 4 °C. The supernatant (cytoplasm) was discarded. The glassy white nuclear pellet was suspended in 500 uL EBX and Triton X-100 was added to a 1% final concentration to lyse the nuclear membrane. The chromatin and nuclear debris were collected by centrifugation (15 K, 10 min, 4 °C). Chromatin was suspended in 50 uL Tris pH 8.0 for Western blot analysis (Chromatin). To each fraction, an equal volume of 2 × SDS-PAGE loading buffer (60 mM Tris pH 6.8, 2% SDS, 10% glycerol, 0.2%bromophenol blue, 200 mM DTT) was added; samples were incubated at 95 °C for 5 min, and were then analyzed by SDS-PAGE and Western blot analyses.

### 4.2. DNA Damage Sensitivity Assays

Serial 10-fold dilutions of logarithmic yeast cells were spotted on fresh synthetic dextrose (SD)-complete (or SD lacking a specific amino acid to preserve the plasmids) plates with or without different concentrations of methyl methanesulfonate (MMS) (Sigma; St. Louis, MO, USA) and incubated at 30 °C for 3 days. MMS plates were freshly prepared, dried in a biological hood, and used the same day.

### 4.3. Western Blot Analysis

Proteins extracted from the fractionation protocol, either from the chromatin fraction or the WCE, were loaded on an acrylamide gel prepared in our lab in 15% acrylamide concentration for PCNA, RPS6 and H3 and on 8% for Srs2. Gels were run for 90 min, half in 140 V in Bio-Rad (Hercules, CA, USA) western-blot construction. Afterwards, the gel was taken apart and proteins were transferred using Bio-Rad (Hercules, CA, USA) transfer construct for 90 min in 400 mA to cellulose membranes. Membranes were incubated for 30 min in 1% skim milk for blocking and afterwards, incubated with primary antibody overnight (anti-PCNA: Sc65598-Santa Cruz, Santa Cruz, CA, USA; anti-Flag: F1804-Sigma-Aldrich, St. Louis, MO, USA; anti-H3: ab1791-abcam, Cambridge, UK; anti-Srs2: sc11991-Santa Cruz, Santa Cruz, CA, USA; anti-RPS6: ab40820-abcam, Cambridge, UK). The following morning, membranes were washed 5 times in TTBS, incubated after each wash for 10 min leading to an hour incubation in a secondary antibody and then, another cycle of 5 washes. In the end, using Thermo Scientific (Waltham, MA, USA) ECL, we exposed the membranes in imager600 and captured pictures of the membrane with different exposures.

### 4.4. Tetrad Dissection

Diploid strains were grown overnight (4–5 mL) in YPD, the cells were spined down (2–3 min) and resuspended in sporulation medium (SPO) in glass tubes. The culture was incubated at 25 °C for 4 days and sporulation was verified under the microscope before continuing to the next step. Spores were treated using beta-glucuronidase and dissected using a micromanipulator. The plates were then grown at 30 degrees for 3 days before being photographed and then replica-plated to various plates to test the relevant markers.

## Figures and Tables

**Figure 1 ijms-25-03359-f001:**
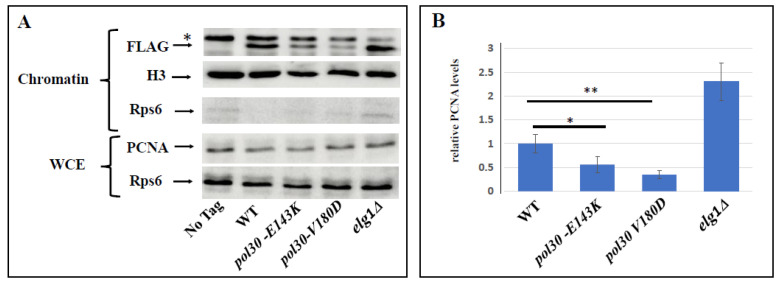
(**A**). The upper three panels are Western blots of the chromatin fraction showing PCNA levels on the chromatin, with histone H3 as a nuclear marker for quantification and Rps6 as a cytoplasmic marker. The lower two panels show PCNA and RPS6 from the whole cell extract (WCE) as a control. The asterisk marks a non-specific band (**B**). Quantitation of this experiment together with 3 additional replicates. *: *p*-value < 0.05; **: *p*-value < 0.005.

**Figure 2 ijms-25-03359-f002:**
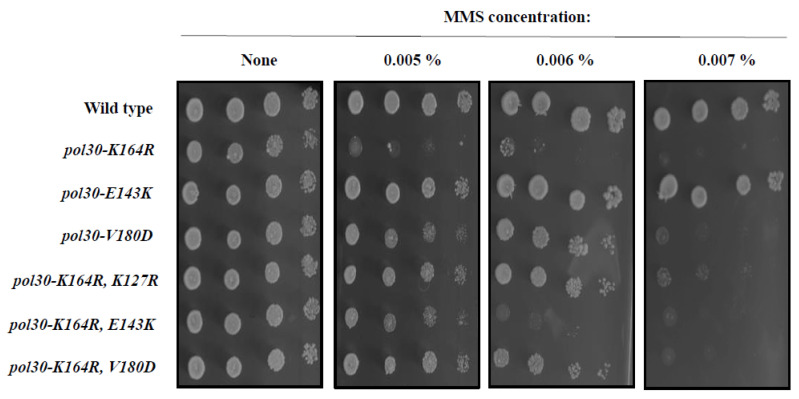
Ten-fold dilution spot assay on plates with increasing amounts of methylmethane sulfonate (MMS). *pol30-V180D* confers mild sensitivity to DNA damage to the same extent as *pol30-K164R*, K127R, while *pol30-E143K* does not cause any apparent DNA damage sensitivity. *The pol30-K164R* mutation greatly sensitizes the cells to DNA damage, and is suppressed by an additional mutation, either *pol30-V180D* or *pol30-K127R*, and to a lesser extent, *pol30-E143K*.

**Figure 3 ijms-25-03359-f003:**
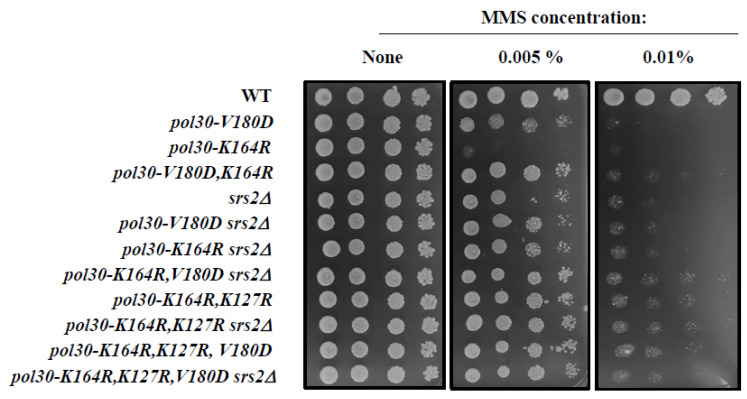
Ten-fold dilution spot assay on plates with increasing amounts of methylmethane sulfonate (MMS). The sensitivity of *pol30-K164R* is suppressed to the same extent by deleting *SRS2*, by introducing pol30-K127R or the DPP mutation *pol30-V180D,* or by their combinations.

**Figure 4 ijms-25-03359-f004:**
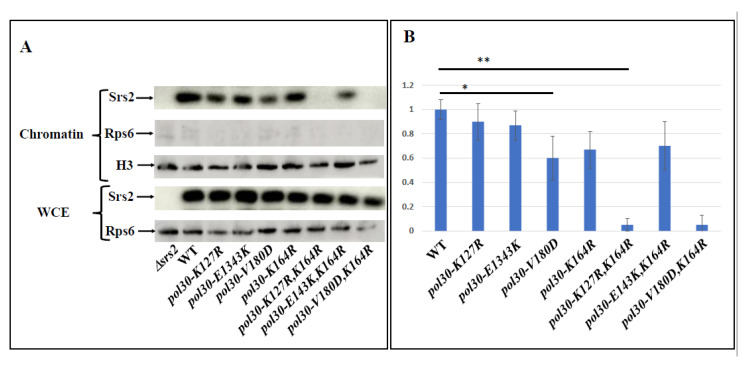
(**A**). The upper three panels are Western blots of the chromatin fraction showing Srs2 levels on the chromatin, with histone H3 as a nuclear marker for quantification and Rps6 as a cytoplasmic marker. The lower two panels show Srs2 and Rps6 from the whole cell extract (WCE) as a control. (**B**). Quantitation of this experiment together with 3 additional replicates. *: *p*-value < 0.05; **: *p*-value < 0.005.

**Figure 5 ijms-25-03359-f005:**
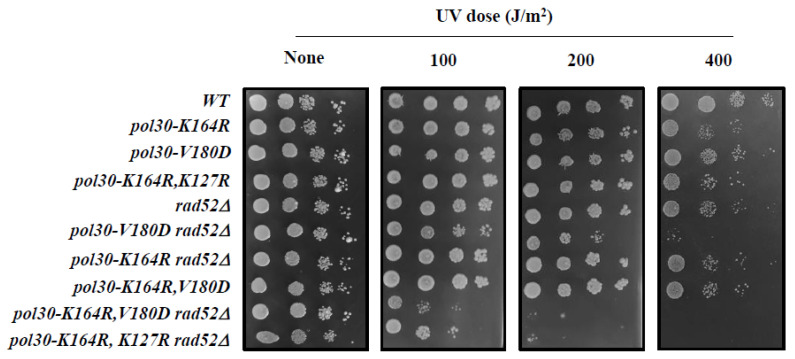
Ten-fold dilution spot assay on plates subjected to increased UV irradiation. The suppression of the DNA damage sensitivity of *pol30-K164R* by *pol30-K127R* and *pol30-V180D* is dependent on the homologous recombination factor Rad52.

**Figure 6 ijms-25-03359-f006:**
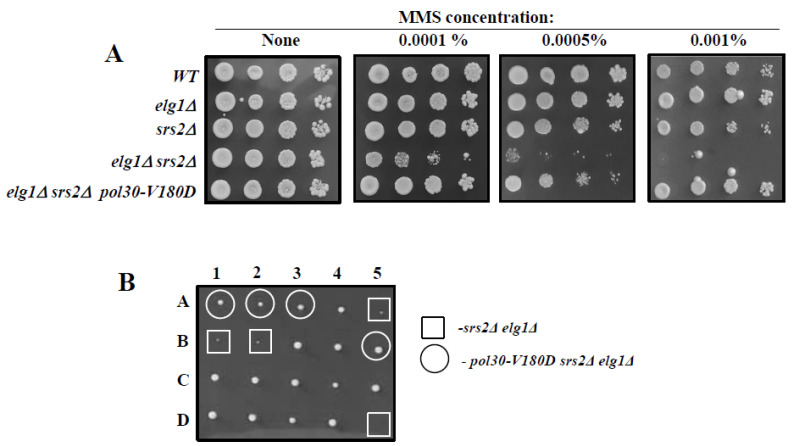
(**A**). Ten-fold dilution spot assay on plates with increasing amounts of methylmethane sulfonate (MMS). (**B**). Tetrad analysis of a diploid created by mating a *pol30-V180D srs2*∆ haploid with an *elg1*∆ haploid of opposite mating type. Whereas the double mutant *elg1*∆ *srs2*∆ is sick (squares), the triple mutant (circles) is not.

**Figure 7 ijms-25-03359-f007:**
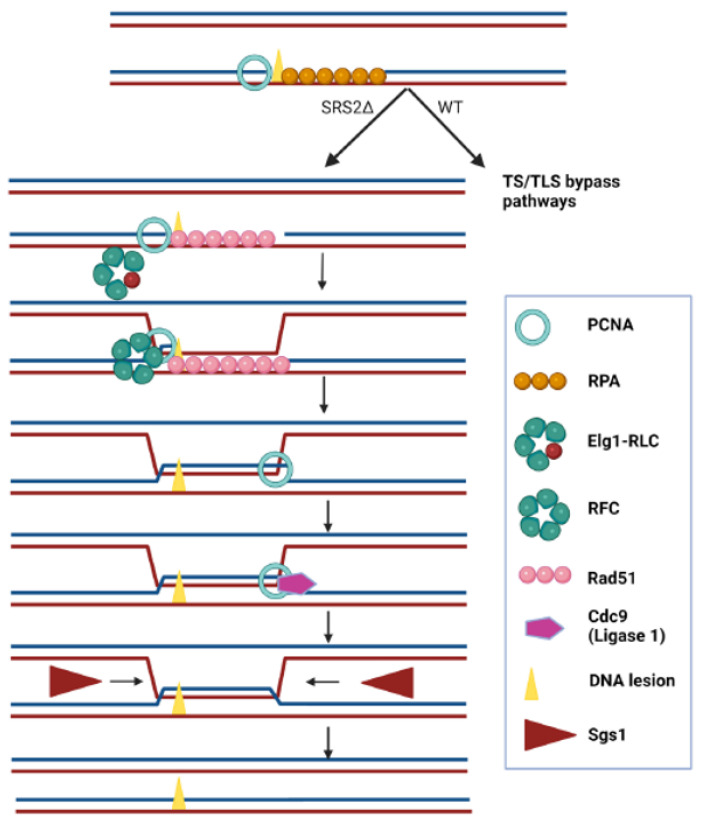
A schematic illustration of the post-replicative bypass of a ssDNA gap left behind the fork in Srs2-deficient cells. A cell in which a replication fork has been arrested due to the presence of a lesion has left a ssDNA gap behind, which is rapidly filled up by RPA. Whereas in w.t. cells ubiquitination of PCNA would promote error-free or error-prone DDT, in the absence of Srs2 RPA is replaced by Rad51, PCNA is unloaded by the Elg1 RLC, and reloaded (by RFC) to allow copying of the information present in the sister chromatid. After ligation (by Cdc9) the intertwined chromatids are resolved by the Sgs1 helicase and the Top3/Rmi1 topoisomerase activity.

**Table 1 ijms-25-03359-t001:** Strains used in this study.

Name	Genotype	Source
18076	*Mat@ ade5-1,lys2::InsEa14,trp1-289,his7-2,leu2-3,112,ura3-52*	[39]
19523	*Mat@ ade5-1,lys2::InsEa14,trp1-289,his7-2,leu2-3,112,ura3-52, Pol30 with C’ terminal flag with KanMX marker*	This study
19787	*MatA ade5-1,lys2::InsEa14,trp1-289,his7-2,leu2-3,112,ura3-52*, *pol30-E143K:LEU22+flag:KanMX*	[30]
19785	*MatA ade5-1,lys2::InsEa14,trp1-289,his7-2,leu2-3,112,ura3-52*, *pol30-V180D:leu2+flag:KanMX*	[30]
19524	*Mat@ ade5-1,lys2::InsEa14,trp1-289,his7-2,leu2-3,112,ura3-52, Elg1::Hyg*, *Pol30 with C’ terminal flag with KanMX marker*	[40]
17822	*Mat@ ade5-1,lys2::InsEa14,trp1-289,his7-2,leu2-3,112,ura3-52, pol30-V180D:LEU2*	[30]
17823	*Mat@ ade5-1,lys2::InsEa14,trp1-289,his7-2,leu2-3,112,ura3-52, pol30-E143K:LEU2*	[30]
19682	*Mat@ ade5-1,lys2::InsEa14,trp1-289,his7-2,leu2-3,112,ura3-52,pol30-K164R*, *K127R:LEU2*	[29]
19808	*MatA,ade5-1,lys2::InsEa14,trp1-289,his7-2,leu2-3,112,ura3-52*, *pol30-K164R::LEU2*	[29]
20573	*MatA*; *ade5-1,lys2::InsEa14,trp1-289,his7-2,leu2-3,112,ura3-52*, *pol30-K164R;V180D:LEU2*	This study
19525	*Mat@ ade5-1,lys2::InsEa14,trp1-289,his7-2,leu2-3,112,ura3-52,pol30-E143K,K164R:LEU2*	This study
20033	*E134 MatA ade5-1,lys2::InsEa14,trp1-289,his7-2,leu2-3,112,ura3-52*, *srs2::KanMX*	[29]
20464	*E134 MatA ade5-1,lys2::InsEa14,trp1-289,his7-2,leu2-3,112,ura3-52*, *srs2::KanMX pol30-V180D: LEU2*	This study
20508	*Mat@ ade5-1,lys2::InsEa14,trp1-289,his7-2,leu2-3,112,ura3-52*, *pol30-K164R:LEU2*, *srs2::KanMX*	This study
20463	*E134 MatA ade5-1,lys2::InsEa14,trp1-289,his7-2,leu2-3,112,ura3-52*, *srs2::KanMX Pol30-V180D,K164R:LEU2*	This study
20461	*E134 matA ade5-1,lys2::InsEa14,trp1-289,his7-2,leu2-3,112,ura3-52*, *Srs2::KanMX*, *pol30-K164R*, *K127R:LEU2*	This study
19678	*Mat@ ade5-1,lys2::InsEa14,trp1-289,his7-2,leu2-3,112,ura3-52*, *pol30-V180D*, *K164R*, *K127R:LEU2-#2*	This study
20462	*E134 MatA de5-1,lys2::InsEa14,trp1-289,his7-2,leu2-3,112,ura3-52*, *Srs2::KanMX pol30-V180D*, *K164R*, *K127R:LEU2*	This study
20609	*E134 MatA ade5-1,lys2::InsEa14,trp1-289,his7-2,leu2-3,112,ura3-52*, *rad52::ura3-1*	This study
20656	*Mat@ ade5-1,lys2::InsEa14,trp1-289,his7-2,leu2-3,112,ura3-52, pol30-V180D:LEU2*, *rad52::URA3*	This study
20589	*Mat@ ade5-1,lys2::InsEa14,trp1-289,his7-2,leu2-3,112,ura3-52*, *pol30-K164R:LEU2*, *rad52::URA3*	This study
20660	*MatA ade5-1,lys2::InsEa14,trp1-289,his7-2,leu2-3,112,ura3-52 Pol30-V180D,K164R:LEU2*, *rad52::URA3*	this study
20708	*MatA; ade5-1,lys2::InsEa14,trp1-289,his7-2,leu2-3,112,ura3-52*, *pol30-K164R,K127R::LEU2 rad52::URA3*	This study
18045	*Mat@ ade5-1,lys2::InsEa14,trp1-289,his7-2,leu2-3,112,ura3-52 elg1::HygMX*	[40]
20545	*Mat@; ade5-1,lys2::InsEa14,trp1-289,his7-2,leu2-3,112,ura3-52*, *elg1::HygMX*; *srs2:KanMX*;	[29]
20544	*Mat@*; *ade5-1,lys2::InsEa14,trp1-289,his7-2,leu2-3,112,ura3-52*, *pol30-V180D:LEU2*; *elg1::HygMX; srs2:KanMX;*	This study
20543	*Diploid*; *ade5-1,lys2::InsEa14,trp1-289,his7-2,leu2-3,112,ura3-52*, *(heterozygous) elg1::HygMX*; *(heterozygous) srs2:KanMX*; *(heterozygous) pol30-V180D:LEU2*	This study
20541	*Diploid; ade5-1,lys2::InsEa14,trp1-289,his7-2,leu2-3,112,ura3-52*, *(heterozygous) elg1::HygMX*; *(heterozygous) srs2:KanMX*; *(heterozygous) Pol30-E143K:LEU2*	This study

## Data Availability

Data is contained within the article.

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
