# Peer review of "PCNA Unloading Is Crucial for the Bypass of DNA Lesions Using Homologous Recombination"

_ijms, 2024, doi:10.3390/ijms25063359_

Round 1

Reviewer 1 Report

Comments and Suggestions for Authors

This manuscript is an interesting yeast genetics study of the intricate DNA damage tolerance pathway choices that occur during DNA replication stress. Using PCNA mutants that associate poorly with chromatin, the authors provide genetic and chromatin fractionation data that support the idea that PCNA unloading is an important step that promotes repair of ssDNA gaps by homologous recombination. The article is generally well-written and the conclusions drawn mostly from genetic crosses and sensitivity assays are reasonable. Provided that a minor experimental point and a number of typos are corrected, this manuscript should be suitable for publication.

Minor point : In Figure 1, the authors present data suggesting that the PCNA trimerization mutants that they use associate poorly with chromatin. However, the WCE blots showing the levels are performed with anti-PCNA antibodies whereas the chromatin fractionations are done with FLAG antibody. To ensure that the total levels of the mutants are similar to that of WT PCNA, the same antibody should be used for both total lysates and chromatin fractionation. If some mutants are not expressed as well as WT PCNA, this should be discussed.

Typos and grammar :

Line 9 cells to progress

Line 110 suppress

Line 159 homologous recombination

In the legend to figure 6 the delta symbols are missing.

Line 187 absent from the fork, the DNA lesion

Line 202 sent

Line 218 A cell

Line 219 in WT cells

Line 248 is Srs2 removed

In table 1 the font size is different for some rows.

Line 279 plasmids

Line 284 Proteins

Line 285 15 % acrylamide

Author Response

Pls see attachment

Reviewer 2 Report

Comments and Suggestions for Authors

In the present study, the authors employed multiple separation-of-function mutants of PCNA to characterize the impact of timely dissociation of PCNA from damaged replication fork. I appreciate the experimental strategies and the results are convincing. I would recommend publication of this article which is highly relevant and provides new insights into the DNA damage response field. Below are the couple of comments the authors should address to further improve the manuscript.

1.      Figure 3. Test the effect of a srs2 mutant lacking SUMOylated PCNA interaction.

2.      To further prove that DPP suppression of PCNA K164R is due to HR restoration, I would recommend examine rad51 immunofluorescence in theses genotypes.

Author Response

Pls see attachment

Round 2

Reviewer 2 Report

Comments and Suggestions for Authors

The authors clarified my comments.